# Artificial Intelligence in Emergency Radiology: Where Are We Going?

**DOI:** 10.3390/diagnostics12123223

**Published:** 2022-12-19

**Authors:** Michaela Cellina, Maurizio Cè, Giovanni Irmici, Velio Ascenti, Elena Caloro, Lorenzo Bianchi, Giuseppe Pellegrino, Natascha D’Amico, Sergio Papa, Gianpaolo Carrafiello

**Affiliations:** 1Radiology Department, Fatebenefratelli Hospital, ASST Fatebenefratelli Sacco, Milano, Piazza Principessa Clotilde 3, 20121 Milan, Italy; 2Postgraduation School in Radiodiagnostics, Università degli Studi di Milano, Via Festa del Perdono, 7, 20122 Milan, Italy; 3Unit of Diagnostic Imaging and Stereotactic Radiosurgery, Centro Diagnostico Italiano, Via Saint Bon 20, 20147 Milan, Italy; 4Radiology Department, Fondazione IRCCS Cà Granda, Policlinico di Milano Ospedale Maggiore, Via Sforza 35, 20122 Milan, Italy

**Keywords:** artificial intelligence, emergency radiology, CAD, deep learning, smart reporting

## Abstract

Emergency Radiology is a unique branch of imaging, as rapidity in the diagnosis and management of different pathologies is essential to saving patients’ lives. Artificial Intelligence (AI) has many potential applications in emergency radiology: firstly, image acquisition can be facilitated by reducing acquisition times through automatic positioning and minimizing artifacts with AI-based reconstruction systems to optimize image quality, even in critical patients; secondly, it enables an efficient workflow (AI algorithms integrated with RIS–PACS workflow), by analyzing the characteristics and images of patients, detecting high-priority examinations and patients with emergent critical findings. Different machine and deep learning algorithms have been trained for the automated detection of different types of emergency disorders (e.g., intracranial hemorrhage, bone fractures, pneumonia), to help radiologists to detect relevant findings. AI-based smart reporting, summarizing patients’ clinical data, and analyzing the grading of the imaging abnormalities, can provide an objective indicator of the disease’s severity, resulting in quick and optimized treatment planning. In this review, we provide an overview of the different AI tools available in emergency radiology, to keep radiologists up to date on the current technological evolution in this field.

## 1. Introduction

Approximately 130 million examinations were performed in Emergency Departments in the United States in 2018, and most patients required imaging exams [1]. In their work routine, emergency radiologists must provide an accurate and timely report, leading to meaningful—sometimes critical—decisions, while facing an everyday-increasing workload, and limited time for image interpretation [2]: in this context, where timing and diagnostic accuracy could significantly impact on the outcome, their performance plays a critical role in patient management, through early diagnosis and appropriate disease characterization [3].

Artificial intelligence (AI) can support emergency radiologists in different tasks, and thus represents a valuable ally in this difficult and stressful environment [4,5]. In the emergency setting, AI can help with patient positioning, image acquisition and reconstruction, worklist prioritization, image interpretation through automatic or assisted anomaly detection, and the creation of a structured report [2] (Figure 1, Table 1).

Not only radiologists and other emergency room professionals would benefit from the integration of AI, but also patients and the entire healthcare system, in terms of quality of care, satisfactory working environments and conditions, and rationalization of healthcare spending [2].

In this narrative review, we provide an overview of AI applications in emergency radiology, to keep radiologists up to date on the developments in this field.

## 2. AI basic Terminology

AI is a generic term that refers to the ability of a machine to exhibit human capabilities, such as reasoning, learning, and problem solving. While human critical and cognitive abilities are irreplaceable, machines have an unrivaled advantage: the ability to analyze huge amounts of data in a very short time [6]. AI-based tools for image analysis underlie the *radiomics* paradigm, synonymous with quantitative imaging, which means considering biomedical images as a set of data, rather than mere pictures [7].

Machine learning (ML) is the core of AI, and concerns the development and tuning of algorithms that demonstrate human-like learning properties for data analysis, but are characterized by performance, robustness, and speed achievable only by a machine. Different ML approaches exploit a large set of candidate maps, to infer the predictions of a quantity of interest (or higher-level parameters) that satisfies some predefined requirements, starting from a given quantity [8,9]: these candidate maps represent different theoretical hypotheses for mapping the relationship between given input and desired output. The fitness of the hypothesis is essentially based on the discrepancy between the prediction and the observed data. The ML model is also expected to reduce the uncertainty of the approximation, as it is exposed to multiple data samples. 

There are four main ML approaches (supervised learning, unsupervised learning, semi-supervised learning, and reinforcement learning), which differ in the level of data pretreatment required, the algorithmic strategies adopted to map the relationships between the data, and the problems they can solve: the first three are the most used in radiology, and each can adapt to different clinical tasks [10].

Supervised learning is the simplest form of ML, and is adequate for very general classification tasks: for example, in pneumonia screening, deciding whether a chest X-ray is normal or not. In this form of ML, the learning phase of the algorithm requires a dataset where each input (independent variable) is classified with the label of the corresponding output (or class). Whereas in classification tasks each variable must be labeled in discrete categories, in linear regression tasks the labeling is expressed by a continuous variable, for example, an outcome, such as disease-free survival. During the training phase, the algorithms learn to infer the relationships between the data, according to a mapping hypothesis (a model), progressively tuning their performance while increasing their exposure to samples. Later, in testing, the model assigns new unlabeled targets to one of the predefined class labels. The overall performance of the algorithm is thus determined by comparing the real labels with the assigned ones [11].

In unsupervised learning algorithms, no label is given in the training dataset, and the machine has to find a hidden structure (if existing) to infer the relationships between the data, or at least a subset of the data. Unsupervised learning can be used to address different problems related to data mining (extracting information from data), such as clustering tasks, where the aim is to divide the dataset into groups based on specific feature characteristics, or association tasks, which aim to find association rules within the dataset [8,9].

Semi-supervised learning falls between unsupervised learning and supervised learning: this approach works primarily on unlabeled data, but it also includes a small amount of labeled data; hence, this type of ML addresses the problem of low data availability, by leveraging the abundant amount of accessible but untagged data (like undiagnosed images), to train accurate classifiers. 

There are many different models for inferring possible relationships between data, and each model features a different design for data representation, analysis, and loss function minimization (a function that describes the approximation towards the desired performance) [8,9]. If we consider the different ML approaches (such as supervised learning) as different strategies for detecting meaningful patterns in data, then each model can be considered as a different tactic. Examples of models in supervised learning are logistic regression, decision trees, and support vector machines. Each model represents a different set of rules for data manipulation, which can be translated into a programming code for automatic computation [11].

While different tactics may be applicable within the same task, some are better performers in a specific context. Among the different models, artificial neural networks are often used to address biomedical imaging tasks. Compared to statistical methods (like logistic regression), artificial neural networks represent a learning paradigm inspired by the workings of the biological neural networks in the human brain.

In recent years, due to the availability of larger datasets and faster computation velocity, particularly complex neural networks, called Convolutional Neural Networks (CNNs), have been developed, paving the way for deep learning (DL) paradigm DL tools are particularly fit to discover intricate patterns in large imaging datasets, beyond the features that could be visually assessed by the radiologist. The depth of a DL model relates to the number of layers within the model [12] with each layer represents an increasing level of abstractionto the point that it is problematic to attribute a meaning to the computations performed by the neural network: this problem is at the basis of the so-called “black box phenomenon”, which represents one of the main challenges in the relationship between human and artificial intelligence (see Section 7 Challenges and perspectives). Overall, DL algorithms showed extraordinary performances in imaging recognition tasks and are at the forefront of the AI-driven revolution in radiology [13,14]. 

## 3. AI in Image Acquisition

Computed tomography (CT) is the first-line technique in emergency radiology, thanks to its wide availability and quick acquisition protocols; however, CT is limited by its exposure to ionizing radiation, although technological advances have allowed for significant dose reduction over the years.

CT scan acquisition algorithms are aimed at improving diagnostic performance, by addressing three main issues: (1) the acquisition of good-quality images; (2) the reduction of acquisition times; and (3) the reduction of the dose, according to the ALARA (as low as reasonably achievable) principle. 

Manual positioning and centering of the patient is a time-consuming and radiotechnologist-dependent activity, essential not only for good diagnostic images, but also for a lower radiation dose.

Due to the presence of the “papillon filter”—a pre-patient attenuator, which blocks the low-energy X-rays directed at the periphery of the analyzed body part—adequate patient positioning in the exact isocenter can better concentrate the X-ray beam [15]: this results in reduced radiation exposure, and better, more focused and less deteriorated images from potential artifacts (e.g., beam hardening).

Several studies have evaluated the feasibility and performances of autopositioning software, which uses anatomical references and scout information to identify the key surface landmark and to define the start and finish positions, in a three-step procedure: (1) identification of the position of the patient and of the various parts of the body; (2) identification of the isocenter; and (3) development of an avatar, using a statistical model adapted to the camera data.

Booji and Saltybaeva evaluated the accuracy of automated patient positioning, using a 3D camera guided by an AI-based algorithm trained with a dataset of images of other subjects, which was capable of recognizing the patient’s surface, and automatically adjusting its position on the table [16,17]. In both studies, significant improvement in patient centering was achieved, compared to manual positioning, with less extreme standard deviations from the ideal isocenter of the patients. More recently, a study by Gang et al., utilizing automated positioning, demonstrated a decrease of 16% in the dosage supplied to the patient, with a saving of 28% of the time [18]. 

The correct selection of the CT study protocol is essential to reducing patient irradiation and preventing crucial information from being lost (causing diagnostic errors or treatment delays). The choice of CT protocol is up to the radiologist, which costs time that could be used for reporting: hence, the need for an AI-based system capable of integrating the patient’s anamnestic data with the purpose of the examination and the conditions of use, in order to suggest the best protocol option. 

The current literature on the subject is limited. MyExam Companion (Siemens Healthcare GmbH, Forchheim, Germany), a recently released data-driven CT scan automation tool, leverages decision trees created from a sizable training dataset of patient characteristics and associated assigned scan procedures, across a broad variety of clinical institutions: this automated approach can include site-specific and regional preferences, and removes technological complexity and mistake risk from the operator, which may thereby result in more uniform image quality [19]. This AI-based strategy could also result in greater procedure uniformity, lowering the margin of radiologists, who sometimes select protocols based on personal preferences and local expertise, rather than an approach based on the data.

Dose reduction is a much-discussed subject, and one of the most promising applications of AI is to improve CT acquisition protocols [20]. Traditional reconstruction techniques, based on filtered back projection algorithms, provide good image quality at the cost of high radiation exposure [21,22]: the more recent Iterative Reconstruction (IR) algorithms, on the contrary, could achieve a lower dose administration while maintaining a diagnostically good quality of image, at the expense of higher reconstruction times [23,24,25].

Fast computation and massive parallelism have enabled the development of AI-based reconstruction algorithms, mostly DL-based, that can achieve dose reduction while still maintaining optimal image quality, compared to traditional filtered back projections and IR algorithms [26].

Park et al. analyzed the performance of a DL reconstruction algorithm, “True Fidelity” (TFI), compared to an IR (“ASIR-V”) system, with regard to the image quality [27]. The algorithms worked at, respectively, an 80–100% blending factor (ASIR-V) and low/medium/high strength (TFI), to reconstruct raw data images of the liver, vessels, and muscle from a 256-slice CT. The images were further validated by two experienced radiologists, and were evaluated on the basis of quantitative (region of interest and noise) and qualitative (tissues inhomogeneity) criteria. The final analysis showed more balanced images, in terms of noise and sharpness, using the DL algorithm compared to the traditional IR system. 

Pediatric radiology departments could significantly benefit from AI-based dose reduction systems. In a recent systematic review, Curtise et al. explored different applications of DL reconstruction algorithms in pediatric patients: their key conclusion was that DL-assisted image reconstructions could provide the radiologist with optimal image quality and an overall dose reduction of 30–80% [28].

DL-based systems are also applicable in other radiological contexts: McLeavy et al. reported their experience with the Advanced intelligent Clear-IQ Engine-AiCE algorithm in an emergency room, highlighting the importance of acquiring low-dose CT, especially in radiosensitive categories, such as fertile women and children [29], demonstrating that with the adoption of this algorithm, the mean dose-length product absorbed by patients in an arterial/venous phase abdomen scan was just 0.8 mSv.

CT acquisition protocols optimization has also been investigated using phantoms: for example, Greffier et al. explored the effect of different levels of DL-based reconstructions on dose reduction and on subjective and objective image quality, compared with the standard IR software routinely installed on CT machines, in a chest imaging setting, using an anthropomorphic phantom [30]. The subjective image quality was independently evaluated by two radiologists, according to different features, such as image noise, image smoothing, vessels-to-fat contrast, border detection, and others. With a “smoother” filter level, the result was a low-noise image, at the same dosage of IR, with improved detectability of simulated chest lesions, but increased image smoothing.

AI tools can also improve artifact reduction algorithms, by replacing the artifact with an approximation inferred by the adjacent zones [31]. 

## 4. Worklist Prioritization

Automatic notification of critical findings is one of the most interesting applications of AI in emergency radiology: with the growing demand for imaging investigations, the delayed communication of salient findings to the referring physician could result in delayed clinical intervention, compromising treatment outcomes, especially in cases requiring immediate action [32].

The priority assigned by the first-line emergency physician usually determines the order in which imaging exams are requested: however, the priority assigned is not always appropriate to the detected anomalies. AI-powered models can pre-identify critical outcomes, and can enable real-time worklist prioritization, reducing report turnaround times and improving clinical pathways: for example, Davis et al. demonstrated a significant decrease in report turnaround time, from 67 to 59 min, and length of stay, from 471 to 425 min [33].

Kao et al. evaluated the effectiveness of Computer-Aided Detection (CAD), connected to the image storage and communication systems and the radiological information system, for identifying, in advance, chest X-rays with anomalies, and bringing them to the radiologist’s attention, as a priority. The sensitivity and specificity of CAD were 0.790 and 0.697, respectively, and its use decreased the turnaround time for abnormal X-rays by 44%, proving the usefulness of CAD systems for first evaluation of the most critical studies [34].

Annarumma et al. created and tested an ML tool based on CNNs, to simulate an automatic triage for adult chest X-rays, according to the urgency of imaging findings, which resulted, for critical studies, in a theoretical reduction in reporting delay, from 11.2 to 2.7 days [5].

AI-based software has also been developed for emergency abdominal imaging, as in a study published by Winkel et al., where the developed CNNs-based algorithm was autonomous in detecting acute pathological abdominal findings with high diagnostic performance, enabling guidance of the radiology workflow toward prioritization of abdominal CT examinations in the acute setting [35].

Emergency findings in neuroradiology have been studied in depth, and several ML-based algorithms have been developed for the automatic detection of brain hemorrhage, mass effect, and hydrocephalus on unenhanced CT examinations, and also for clinical workflow integration [36,37,38]. Arbabshirani and his colleagues obtained interesting results using an ML algorithm trained on 37,074 head CT scans, improving the prioritization of worklists, and reducing the time taken to diagnose intracranial hemorrhage [39].

## 5. Automatic Detection

### 5.1. Stroke

Head trauma, intracranial hemorrhage, and ischemic stroke are medical emergencies with a high mortality rate and severe neurological outcomes worldwide: therefore, rapid identification through neuroimaging evaluation is essential, to ensure the best management [39,40,41]. AI-based tools can enable the automatic identification of different neurological emergencies, so as to improve the timing and selection of the most appropriate therapy [42].

In the last decade, AI-based software has been developed for emergency neuroimaging, demonstrating high performance and expert-level accuracy for the diagnosis of intracranial hemorrhage and chronic cerebral microbleeds on non-contrast CT. Matsoukas et al., for example, reported overall sensitivity, specificity, and accuracy of 92.06%, 93.54%, and 93.46%, respectively, for intracranial hemorrhage detection, and 91.6%, 93.9%, and 92.7% for cerebral microbleeds [43].

Rava et al. retrospectively assessed the AI-based Canon ^AUTO^Stroke solution for the detection of intracranial hemorrhages in patients who presented with stroke-like symptoms: automated analysis of CT scans demonstrated 93% sensitivity, 93% specificity, a positive predicting value of 85%, and a negative predicting value of 98% for patients with intracranial hemorrhage; 95% of positive patients were correctly triaged, whereas 88.2% of negative cases were correctly classified as negative for intracranial hemorrhage [44].

In a study by Ginat et al., the accuracy of DL software for automated acute intracranial hemorrhage detection depended on the timing of the CT scan (initial versus follow-up) and the clinical context (emergency department, inpatient or outpatient) [45].

AI can also analyze CT angiography images for large vessel occlusions. McLouth et al. evaluated an ML model for the detection of intracranial hemorrhages, and reported 98.1% accuracy, 98.1% sensitivity, and 98.2% specificity in the diagnosing of large vessel occlusion. A random subset of 55 cases was also assessed for the detection of the site of occlusion—including the distal internal carotid artery, the middle cerebral artery M1 segment, the proximal middle cerebral artery M2 segment, and the distal middle cerebral artery M2 segment—with 97.0% accuracy, 94.3% sensitivity, and 97.4% specificity [46].

Future applications include the automatic assessment of collateral circulation in CT angiography, as a prognostic factor in acute ischemic stroke [47].

### 5.2. Trauma and Bone Fractures

Musculoskeletal emergencies are frequent causes of access to emergency departments, with traumas to the extremities accounting for 50% of all nonfatal injury costs globally [48]; however, experienced radiologists’ mistake reporting rate is estimated at approximately 4%, resulting in delayed diagnosis and increased morbidity [49]. AI can serve as a support tool for the radiologist, while maintaining or improving diagnostic accuracy, despite increased workload [50,51,52].

AI systems are most commonly used in fracture detection. Krogue et al. reported the results of a DL-based system to detect and classify bone fractures, which improved residents’ performance [53]. Another study investigated a DenseNet 121-layer network with three prediction classes—including normal—to identify and characterize proximal femur fractures: it reported an area under the curve of 0.98, accuracy of 91%, sensitivity and specificity of 98% and 84%, respectively, and an F1 score of 0.916 [54]. Jones et al. developed a DL system that was tested on X-ray examinations of adult patients, involving 16 anatomical regions and different types of fractures, observing that the AI could accurately replicate the skills of orthopedic surgeons and radiologists in identifying fractures, with an overall AUC of the DL system of 0.974 [55]. 

As plain radiographs are the first-line examination in the trauma setting, most studies on AI in musculoskeletal emergencies have focused on them; however, initial evidence is available concerning CT scan detection of fractures of the spine, calcaneus, and femoral neck, with promising results [49].

Less research is available on AI in magnetic resonance (MR) imaging for fracture detection, due to its limited availability in the emergency setting; however, some studies have investigated AI in magnetic resonance imaging of ligamentous, cartilage, and meniscal lesions. 

One hundred sagittal MR knee images, from patients with and without anterior cruciate ligament lesions confirmed by arthroscopy, were used by Minamoto et al. to train a CNN, whose performance was compared with the evaluation of twelve radiologists: the AI system demonstrated sensitivity of 91%, specificity of 86%, accuracy of 88.5%, a positive predictive value of 87.0%, and a negative predictive value of 91.0% [56].

In another study, a 14-layer ResNet-14 architecture of CNN was developed, using 917 knee sagittal MR images for the early detection of cruciate ligament tears, with accuracy of 92%, sensitivity of 91%, and specificity of 94% [57].

Bien et al. used a dataset comprising 1370 knee magnetic resonance examinations, performed at Stanford University Medical Center, to develop a CNN, called “MRNet”, to classify knee injuries. The CNN achieved an area under the curve of 0.965 for anterior cruciate ligament lesions and 0.847 for meniscal tears, when compared to three musculoskeletal radiologists [58].

In another study, a mask region-based CNN demonstrated an area under the curve of 0.906 in the identification of meniscal tears [59].

A DL-based system for detecting cartilage lesions, that was tested on 175 patients, and compared to interpretation by musculoskeletal radiologists, demonstrated high diagnostic accuracy [60]. Similar results were obtained by Roblot et al., who used a CNN of 1123 participants to identify meniscal tears, yielding an area under the curve of 0.94 [61]. When Chang et al. employed CNN U-Net to distinguish between 130 cases with and 130 without torn anterior cruciate ligaments, the DL showed an area under the curve of 0.97, with 100% sensitivity and 93% specificity [62].

One of the more recent applications in the field of fractures is for the assessment of osteoporotic fractures. Ferzi et al. developed a model to predict and diagnose spinal fragility fractures in patients with osteoporosis, and demonstrated that the best algorithms for predicting osteoporotic fractures with an F1 value of 0.64 are those supplemented with RUS (an algorithm that removes scatter from the data distribution), logistic regression, and the linear discriminant [63].

Arpitha et al. developed a CAD for the detection of lumbar vertebral compression fracture and automatic classification into “malignant” and “benign” [64].

A recent systematic review, including 42 studies on the topic of AI for fracture detection by any radiological techniques, showed no statistically significant differences between the performance of AI and clinicians [65], suggesting that AI technology represents a promising tool for fracture diagnosis in future clinical practice.

### 5.3. Abdominal Emergencies

There is growing interest in the effectiveness of AI in abdominal emergency imaging; however, compared to other contexts, such as skeletal fracture detection, abdominal imaging presents more demanding and specific challenges, related to the variability of abdominal anatomy and the complexity of imaging features. For these reasons, the application of AI in the emergency context is still in its infancy, with only preliminary studies testing its possible application, and no solid literature yet to support its widespread integration into clinical routine; however, initial evidence shows that AI can be useful as a second reader, increasing radiologist diagnostic confidence and decreasing the time to diagnosis [66]. 

#### 5.3.1. Abdominal Trauma

When discussing abdominal trauma, the detection of free fluid in the abdominal cavity is critical to determining the patient’s correct therapeutic approach: misdiagnosis could have dramatic consequences [67,68,69]; a focused Assessment with Sonography for Trauma (FAST) is usually the first imaging approach to diagnosing hemoperitoneum, with particular attention paid to the Morison pouch [70]. 

Cheng et al. investigated the performance of ultrasound detection of abdominal free fluid in the Morison pouch, using the Residual Networks 50-Version 2 (ResNet50-V2) DL algorithm: they gathered 324 patient records for the training model, 36 patient records for validation, and 36 more for testing. The model performance for ascites prediction was 0.961 for accuracy, 0.976 for sensitivity, 0.947 for specificity in the validation set, and 0.967, 0.985, and 0.913 in the test set, respectively. The human interpretation result was not significantly different from the DL model (*p* = 0.570) [71]. 

Although ultrasound detection of abdominal free fluid may seem like a simple task for experienced radiologists, the authors believe that this AI system can help inexperienced sonographers and paramedics in their interpretation of FAST assessment, improving the outcome of blunt abdominal trauma.

Another interesting study by Drezin et al. evaluated AI in blunt hepatic trauma, to increase the diagnostic capacity of CT in detecting vascular damage [72]. Liver injuries are frequent findings in patients who undergo an emergency CT scan for blunt trauma, and contrast extravasation represents the most direct sign of arterial bleeding, but may be discordant with angiographic data, often underestimating the extent of bleeding [73]. 

In this study, the authors re-trained and validated previously described DL algorithms, to obtain the automated measurement of the liver parenchymal disruption index (auto-LPDI) on 73 patients from two centers, submitted to catheter-directed hepatic angiography post-CT [72]. The results showed that a decision tree based on auto-LPDI and volumetric contrast extravasation measurements had the highest accuracy (0.84), and was a significant improvement over contrast extravasation assessment alone (0.68), with the demonstration that auto-LPDI was a significant independent predictor of major hepatic arterial injuries in these patients.

#### 5.3.2. Small Bowel Occlusion

Small bowel occlusion is a partial or complete blockage of the small intestine, representing a common cause of acute abdominal pain, usually investigated with plain abdominal radiography as a first-level examination [74].

Radiological findings specific to small bowel occlusion include multiple air–fluid levels [75]; AI-assisted detection may be helpful for guiding junior radiologists in diagnosing this pathology [76].

In a recent paper, Kim et al. investigated the feasibility of a DL model for the automated identification of small bowel obstruction: they collected 990 plain abdominal radiographs (445 with normal findings and 445 with small bowel obstruction), and used the data to develop a predictive model comprising an ensemble of five CNNs: they reported a good diagnostic performance, with an area under the curve of 0.961, 91% sensitivity, and 93% specificity [77].

In another study, Goyal et al. developed an ML-based prediction model for closed-loop small bowel obstruction, which integrated CT images and clinical findings, based on the retrospective evaluation of 223 patients who underwent surgery for suspected closed-loop small bowel occlusion [78]. Two radiologists individually analyzed CT scans, searching for typical findings, blinded to the surgically confirmed diagnosis or clinical parameters: diagnosis of small bowel occlusion at surgery represented the reference standard. Clinical data extracted from the electronic medical records included age, sex, history of abdominal/pelvic surgery and of intra-abdominal cancer, and serum lactate. The algorithm combining imaging and clinical findings yielded an area under the curve of 0.73, sensitivity of 0.72, specificity of 0.8, and accuracy of 0.73.

#### 5.3.3. Intussusception

Another important gastrointestinal tract emergency is bowel intussusception, a pathology that occurs when one segment of the bowel is pulled into itself or into another adjacent bowel loop by peristalsis [79], representing a cause of acute abdomen in children, which needs quick clinical and instrumental assessment. 

Generally, an X-ray of the abdomen does not allow a reliable diagnosis of intussusception, and ultrasound represents the first-level examination [80]. Ultrasound, however, is a time-consuming method, highly dependent on the available equipment and on the experience of the operator [81]. Given the lack of specific signs and symptoms in many cases, the availability of a supporting tool would be ideal.

Recent studies have focused on the implementation of DL algorithms in abdominal X-rays, suggesting that AI integration may bring added value to image interpretation. In a multicenter retrospective study, Kwon et al. evaluated a deep CNNs algorithm to detect intussusception in children in plain abdominal X-rays: they collected 1449 images of intussusception in children ≤6 years old, and 9935 images of patients without intussusception from three centers [82]; using a Single Shot MultiBox Detector for abdominal detection, and a ResNet for intussusception classification, Kwon et al. analyzed the diagnostic performance with internal and external validation tests, achieving high diagnostic accuracy [82].

### 5.4. Chest Emergencies

Chest imaging is a critical part of emergency radiology, and AI can play an important role in helping radiologists make the correct diagnosis quickly: most studies have focused on two topics—pulmonary embolism and pneumonia.

#### 5.4.1. Pulmonary Embolism

Pulmonary embolism is a serious and potentially fatal condition: it is the occlusion of the pulmonary arteries or their branches, due to a thrombus, originated in the systemic venous circulation, that causes an impairment of parenchymal perfusion.

The gold standard technique for diagnosis of pulmonary embolism is CT pulmonary angiography (CTPA) [83,84].

Analysis of CTPA scans for pulmonary embolism is a challenging task, with a high risk of false positives, on account of the presence of artifacts and false negatives attributable to the lack of detection of small thrombi located in the distal bronchi; it is also a time-consuming task, due to the complex bronchial anatomy, with the consequent possibility of diagnostic errors and therapeutic delays [85].

Nowadays, there is a growing interest in developing reliable and fast AI-based systems to assist radiologists as second readers. Cheik et al. evaluated the diagnostic performances of an AI-powered algorithm for the automatic detection of pulmonary embolism in the CTPAs of 1202 patients, and compared the results with those of emergency radiologists in routine clinical practice [86]. 

The AI algorithm detected 219 suspicious pulmonary embolisms, of which 176 were true embolisms, including 19 missed by radiologists; moreover, the highest sensitivity and negative predictive values were obtained by the AI (92.6% versus 90%, and 98.6% versus 98.1%, respectively), while the highest specificity and positive predictive values were obtained by the radiologists (99.1% versus 95.8%, and 95% versus 80.4%, respectively). Because of the high negative predictive value, the authors believe that AI tools for pulmonary embolism detection could help radiologists in emergency settings.

Batra et al. studied the diagnostic performance of a commercial AI algorithm for the detection of incidental pulmonary embolism (iPE) on conventional contrast-enhanced chest CT examinations performed for different reasons^88^. They retrospectively evaluated 3003 examinations, of which 40 had iPE [87]. AI detected four embolisms missed in clinical reports, while clinical reports described seven embolisms missed by AI. Compared to radiologists, AI had a lower positive predictive value (86.8% versus 97.3%; *p* = 0.03) and specificity (99.8% vs. 100.0%; *p* = 0.045), while differences in sensitivity and negative predictive value were not significant.

In a recent meta-analysis, Soffer et al. evaluated a total of 36.847 CTPA in seven studies, to establish the diagnostic performance of DL-based algorithms in the diagnosis of pulmonary embolism on CTPA: they found that the pooled sensitivity and specificity were 0.88 and 0.86, respectively, suggesting that DL models can detect pulmonary embolism on CTPA with satisfactory sensitivity and an acceptable number of false positive cases [88].

Although there are high expectations regarding the use of AI in the detection of pulmonary embolism, and the majority of studies conclude in favor of its use in clinical practice, others have expressed reservations.

For example, Müller-Peltzer et al. conducted a retrospective study to evaluate the diagnostic accuracy of a commercially available AI-based tool (syngo.via VB20, Siemens Healthineers, Erlangen, Germany), which included 1229 CTPA scans performed over a 36-month period, and was analyzed by two expert radiologists taken as a reference standard [89]. The radiologists’ analysis detected a total of 504 emboli pulmonary in 182/1229 patients, while the CAD algorithm reported 3331 emboli; within 504 pulmonary embolisms identified by the radiologists, the CAD tool identified only 258 embolisms, corresponding to a sensitivity of 51.2% (95% CI: 0.47–0.56), with the remaining findings (3073 embolisms, 92%) being false-positive—which means, on average, that there were 2.5 ± 2.54 false-positive findings per patient. 

These results should be taken as a cautionary note against overreliance on these tools.

#### 5.4.2. Pneumonia

“Pneumonia” is a commonly used generic term that includes different clinical and radiographic features, depending on the etiological agent. Recently, given the global spread of the COVID-19 pandemic, the literature has mainly focused on the detection of COVID-19 infections [90]. 

CT findings of COVID-19 pneumonia include ground-glass opacities and peripheral alveolar consolidations, but these findings have a low specificity because they are also seen in other non-COVID pneumonia [91]. AI can help distinguish between COVID-19 and non-COVID pneumonia. Bai et al. evaluated an AI algorithm to differentiate COVID-19 and other pneumonia on chest CT, and compared the results with radiologists’ performance without and with AI assistance. They collected 521 chest CT scans of patients with a swab diagnosis of COVID-19, and 665 chest CTs of patients with non-COVID-19 pneumonia, and imaging evidence of pneumonia [92]. Then, six radiologists, with at least 10 years of experience, reviewed the test set indicating COVID or non-COVID pneumonia, two times—the first time without AI, and the second time with the prediction from AI: their analysis showed not only that their model, compared to radiologists, achieved a higher test accuracy (96% vs. 85%, *p* < 0.001), sensitivity (95% vs. 79%, *p* < 0.001), and specificity (96% vs. 88%, *p* = 0.002) but also that, assisted by the models’ probabilities, the radiologists achieved a higher test accuracy (90% vs. 85%, ∆ = 5, *p* < 0.001), sensitivity (88% vs. 79%, ∆ = 9, *p* < 0.001), and specificity (91% vs. 88%, ∆ = 3, *p* = 0.001).

In another study from 2017, before the COVID pandemic, Rajpurkar et al. developed a 121-layer convolutional neural network algorithm (CheXNet) trained on the chestX-ray14 dataset, and evaluated its diagnostic performance in comparison to four practicing academic radiologists with at least 4 years of experience in detecting pneumonia [93]: they collected 420 frontal X-rays, and found that CheXNet achieved an F1 score of 0.435 (95% CI 0.387, 0.481), which was higher than the radiologist average of 0.387 (95% CI 0.330, 0.442).

The main characteristics of the above studies are summed up in Table 2.

## 6. Smart Reporting

Structured radiological reporting is essential for ensuring completeness and comparability, and for reducing ambiguity, while improving communication with other clinicians [94]. Due to technological issues and a lack of interaction with the current reporting systems, the use of structured reports is still not common in clinical practice [95]. Natural language processing, often known as text mining, is an AI-based task that combines linguistics, statistics, and DL-models to evaluate free-form text [96,97]: this procedure results in each part having a structured format, with a predetermined arrangement and nomenclature [95].

The first phase of natural language processing is feature extraction, which consists of segmentation, sentence splitting, and tokenization [96]: the goal is not to create a simple quantitative report like the blood analysis model, but rather a truly informative report, combining quantitative parameters with radiologist findings. The process of separating radiological reports into their component parts is referred to as “segmentation.” The terms “sentence splitting” and “tokenization” are used to describe this process [97]. The semantic analysis desired output is to output each individual notion as a single item in a structured format. The primary natural language processing technologies used for these tasks are pattern matching and language analysis: the latter, which is more complicated, may highlight which concepts are quoted in the text, and how each concept is related to other concepts; the former involves matching a string of letters. NegEx is a pattern-matching tool that finds negational terms within a short string of words, before and after a given concept, such as “no” or “absent” [98].

The previously described processes determined the natural language processing features. To address this issue, we must now extract data from these features. ML and DL may be able to handle textual features. ML previously required training labeled data to create a relationship between the extracted features and a predetermined class: in this situation, the classifier’s performance is highly dependent on the training set [99]. Recurrent neural networks are DL models used to process natural language processing characteristics, because they handle sequential information, and are appropriate in NLP, as sentences are collections of words [96].

The main applications of natural language processing include the identification and classification of results, the automatic elaboration of diagnostic recommendations, and diagnostic surveillance and identification of cases for potential research studies [96].

The integration of these AI systems into RIS/PACS systems is an essential point for the widespread diffusion of AI-based structured reports [100]. 

## 7. Challenges and Perspectives

Radiomics represents a turning point for radiology, but some challenges concerning the development and application of AI-based tools in clinical practice still need to be addressed [101].

AI requires a multidisciplinary team effort, large amounts of high-quality data, and a rigorous workflow. These limitations initially frustrated researchers’ high expectations, leading to what has been commonly referred to as the “winter of artificial intelligence”, a period of reduced funding and interest in AI research [102]; however, in recent years, a series of methodological acquisitions have given new impetus to the enormous potential impact of AI in biomedical imaging, particularly the emergency setting, where it is necessary to complete the transition from a decision-making process based on subjective evaluations to one based on data.

The development of AI-based tools requires multidisciplinary knowledge that cannot be managed by a single radiologist: this can be an obstacle for small groups and research institutions that lack the financial resources to recruit non-medical professionals exclusively for research or support activities.

The availability of large amounts of data is the primary constraint on the development of ML models, which must be trained, validated, and tested on large datasets: in recent years, the growing popularity of open-source image repositories has partially addressed this problem, encouraging collaborative research between different institutions, which also represents a stimulus to standardizing acquisition protocols [103].

The problem of data quality is another threat to the development of AI-based tools, due to the heterogeneity of image acquisition protocols, manufacturers, and post-processing algorithms currently adopted in clinical routine. High-quality images are needed, and are possibly acquired using the same machine, and with a high signal-to-noise ratio. If the quality of the data is poor, the quality of the resulting models is also poor. Several methods have been developed for data harmonization, which could significantly impact on radiomic features’ reproducibility [104,105,106,107].

Another key issue concerns the interpretability of results from AI-based models, in relation to the so-called “black box” phenomenon. Accuracy and interpretability may not be on the same level: for example, “decision tree” models enjoy high interpretability, because it is easy to understand the algorithm inner workings, and to attribute a specific meaning to their operation; however, they can be inaccurate for complex tasks. On the other hand, deep CNN shows excellent performance in image analysis, at the expense of less interpretability. A proper understanding of how predictions are made is a legitimate concern for clinicians, especially in emergency departments in which a wrong decision may lead to disastrous consequences [108,109,110].

Model validation is imperative to ensuring its applicability, as the training dataset represents a sample of the population that may be more or less representative of the population itself. In the absence of validation, it would not be surprising to see the performance of the model decrease and even collapse in a real-world setting. Validation is a complex step that must occupy the right place in the radiomics workflow, and choosing the right validation approach (e.g., random sampling, k-fold cross-validation, bootstrap) depends on several factors, including the type of model, and the quantity and quality of data [8,9].

Medical imaging research is facing a proliferation of studies proposing new AI-powered diagnostic and predictive tools, and promising outstanding performances. Relatively few studies have focused on evaluating the applicability of these models, and what concrete benefits they would bring to clinical practice in the real-world setting [110,111,112]: the results can be surprisingly disappointing, when such models are tested in different cohorts of patients or institutions. In the emergency setting, for example, disappointing results achieved by some software should warn radiologists against placing excessive reliance on these tools^90^.

The fact that AI-based tools underperformed in the real-world setting could address a general concern of emergency radiologists about the potential threat posed by AI to their profession [113]; However, on deeper analysis, including demystification of AI, it is generally accepted that the role of the radiologist is unlikely to be seriously threatened by AI, at least in the coming decades. Human decision-makers have other tasks that go beyond the interpretation of the images, and include the choice of the appropriate diagnostic examination, the analysis of complex cases that require multimodal integration, the relationship with colleagues, the interpretation of outcomes in light of the clinical context, patient communication, and interventional procedures [114] The most likely prospect is that AI cannot replace radiologists; however, “radiologists who use AI will replace those who don’t”, as AI can facilitate everyday tasks performed by radiologists, and can increase diagnostic performance [115].

It is likely that in the next few years there will be a redefinition of the role of the radiologist, which will entail profitable cross-fertilization between different professional roles: only in this way can the challenges outlined above be effectively addressed.

Emergency departments are an important proving ground for testing the applicability and benefits of AI-powered tools. In an emergency context, artificial intelligence tools could support radiologists in long and highly repetitive tasks, reducing diagnostic errors where the workload, expectations, and risk of making errors are high [116]. Successful integration of AI could potentially free up resources to be devoted to other activities, such as patient communication, that have been sacrificed too much, due to increased workload [117]

Several studies have shown that the main merits of currently available commercial software are high sensitivity and negative predictive value—for example, in pulmonary embolism detection [89]: such software could be suitably integrated in the pre-screening phase, leaving the task of confirming the diagnosis to the expert radiologist.

The application of AI in medicine responds to diagnostic and therapeutic problems; however, AI should not be considered only as a tool, but as a leading actor in a complex socio-technical system: a question, therefore, arises as to the trustworthiness of AI, understood as a property of a socio-technical system that satisfies some general criteria from a legal, ethical, and technical point of view [118].

On 8 April 2019, the High-Level Expert Group on AI set up by the European Commission presented a programmatic document which should be considered as a reference guideline to handle AI integration in various fields, including healthcare, aiming to increase individual and collective human well-being [119]. This document was followed by the White Paper on Artificial Intelligence in 2020 [119]. According to these documents, reliable AI should be: (1) lawful—respecting all applicable laws and regulations, including privacy requirements; (2) ethical—respecting ethical principles and values; and (3) robust—both from a technical point of view, and considering its social environment and explainability [119].

## 8. Conclusions

We present an overview of AI applications for emergency radiology. Most of these applications are still in the research phase: however, we believe that, in the future, at least some of them will be ready for widespread adoption in everyday clinical practice.

AI has potential to improve healthcare delivery, including medical imaging. Radiologists should be aware of the available AI tools, to apply them when needed, so as to improve their diagnostic accuracy and management of emergent cases.

## Figures and Tables

**Figure 1 diagnostics-12-03223-f001:**
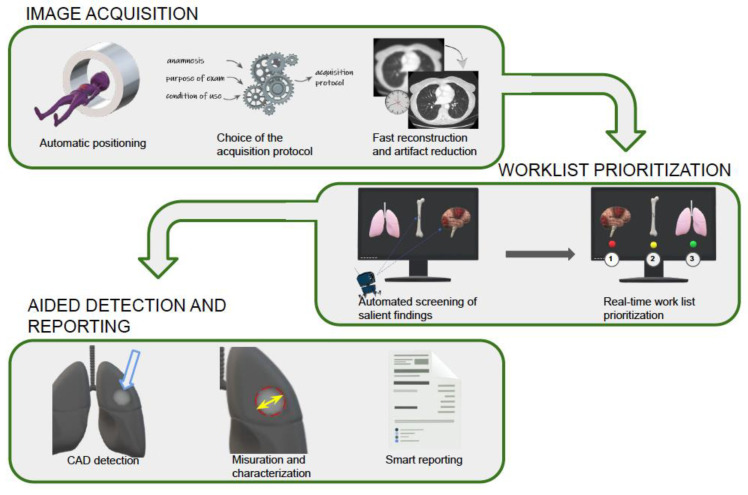
Graphic explanation of the functions that AI tools can carry out in emergency radiology.

**Table 1 diagnostics-12-03223-t001:** This table shows the main characteristics of different types of machine learning approaches.

Type of ML	Mechanism	Type of Data Provided	Tasks for which the Mechanism can be used	Examples of Models
Supervised learning (SL)	The algorithm, provided with tuples (x,y) of input (labeled) and output (unlabeled), infers the relations that map the data.	Labeled data	Classification task; Regression task	Logistic regression Decision Tree Random Forest Support Vector Machines Artificial Neural Networks
Unsupervised learning (UL)	The algorithm exhibits self-organization, to capture hidden patterns in data.	Unlabeled data	Clustering Association; Anomalies detection	Hierarchical clustering K-mean Artificial Neural Networks
Semi-supervised learning (SSL)	The algorithm is placed between unsupervised learning (with no labeled training data) and supervised learning (with only labeled training data).	Mostly unlabeled data, with a small amount of labeled data.	Transductive task (infer the correct labels for the given unlabeled data) or inductive tasks (infer the correct mapping from x to y).	Generative model Self-training model Co-training model Transductive model Graph-based model
Reinforcement learning (RL)	The algorithm is created with a goal and a set of rules. The algorithm tends to maximize the "reward function" or reinforcement signals, to achieve the goal.	Not needing labeled input/output pairs to be presented; only a numerical performance score is given as guidance.	Good for modeling complex-task decision-making processes, such as economics and game theory under bounded rationality.	Monte Carlo methods Q-learning State–action–reward–state–action methods

**Table 2 diagnostics-12-03223-t002:** This table sums up the main results from the studies investigating the different applications of AI tools for automatic disease detection.

Section	Authors	Main Application	Technique	Findings
Neuroradiology	Matsoukas et al. [43]	Intracranial hemorrhages	CT	Sensitivity, specificity, and accuracy of 92.06%, 93.54%, and 93.46%.
		Cerebral microbleeds	CT	Sensitivity, specificity, and accuracy of 91.6%, 93.9%, and 92.7%.
	Rava et al. [44]	Intracranial hemorrhages	CT	Sensitivity of 93%, specificity of 93%, a positive predicting value of 85%, and a negative predicting value of 98%.
	McLouth et al. [46]	Large vessel occlusion	CT	Accuracy of 98.1%, sensitivity of 98.1%, specificity of 98.2%.
MSK	Cheng et al. [54]	Femoral fractures detection	X-ray	AUC of 0.98, accuracy of 91%, sensitivity of 98%, specificity of 84%, and an F1 score of 0.916.
	Jones et al. [55]	Fractures detection in 16 anatomical regions.	X-ray	AUC of 0.974, sensitivity of 95.2%, specificity of 81.3%, a positive predictive value (PPV) of 47.4%, and a negative predictive value (NPV) of 99.0%.
	Minamoto et al. [56]	Anterior Cruciate Ligament lesion	MRI	Ssensitivity of 91%, specificity of 86%, accuracy of 88.5%, a positive predictive value of 87.0%, and a negative predictive value of 91.0%.
	Bien et al. [58]	Anterior Cruciate Ligament lesion	MRI	AUC of 0.965, when compared to three musculoskeletal radiologists.
		Meniscal tears	MRI	AUC of 0.965, when compared to three musculoskeletal radiologists.
	Liu et al. [60]	Meniscal tears	MRI	Sensitivity and sensibility of 84.1% and 85.2%, respectively, for evaluation 1, and of 80.5% and 87.9%, respectively, for evaluation 2. Areas under the ROC curve were 0.917 and 0.914 for evaluations 1 and 2, respectively.
	Roblot et al. [61]	Meniscal tears	MRI	AUC of 0.92 for the detection of the position of the two meniscal horns, of 0.94 for the presence of a meniscal tear, of 0.83 for determining the orientation of the tear, and a final weighted AUC of 0.90.
Abdominal	Cheng et al. [71]	Ascites in the Morison pouch	Ultrasound	0.961 for accuracy, 0.976 for sensitivity, 0.947 for specificity in the validation set, and 0.967, 0.985, and 0.913 in the test set, respectively.
	Drezin et al. [72]	Measurement of the liver parenchymal disruption index	CT	Accuracy of 0.84
	Kim et al. [77]	Small bowel occlusion	X-ray	AUC of 0.961, sensitivity of 91%, specificity of 93%.
	Goyal et al. [78]	Closed-loop small bowel occlusion	CT	AUC of 0.73, sensitivity of 0.72, specificity of 0.8, accuracy of 0.73.
Chest	Cheik et al. [86]	Pulmonary embolism	CT	The AI had the best sensitivity and negative predictive values (92.6% vs. 90%, and 98.6% vs. 98.1%, respectively), whereas radiologists had the highest specificity and positive predictive values (99.1% vs. 95.8%, and 95% vs. 80.4%, respectively).
	Batra et al. [87]	Incidental pulmonary embolism	CT	AI had a lower positive predictive value (86.8% versus 97.3%, *p* = 0.03) and specificity (99.8% vs. 100.0%, *p* = 0.045) vs. radiologists.
	Soffer et al. [88]	Pulmonary embolism	CT	Sensitivity and specificity were 0.88 and 0.86, respectively.
	Xiong et al. [92]	COVID-19 pneumonia	CT	Accuracy of 96%, sensitivity of 95%, and specificity of 96%.
	Rajpurkar et al. [93]	Pneumonia	X-ray	F1 score of 0.435.

AI = artificial intelligence; AUC = area under the curve.

## Data Availability

Not applicable. This is a review article. We did not use any collected data.

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
