# Peer review of "Artificial Intelligence in Emergency Radiology: Where Are We Going?"

_diagnostics, 2022, doi:10.3390/diagnostics12123223_

Round 1
Reviewer 1 Report
The authors have put in a lot of effort to write a review article regarding “Artificial Intelligence in Emergency Radiology”. The topic itself is appealing to readers. However, the content is lengthy and difficult to understand.
Comments:
1. I suggest that the authors can present their review in a more succinct style.
2. There are too many abbreviations in the text and can be misleading or misrepresented especially if they are not checked properly. Some abbreviations are even not spelled out for their first appearance in the text. I strongly recommend that the authors can reduce their frequency.
3. Page 3, line 103, “….data, hence this type of ML. This type of ML addresses…” is not easy to understand.
4. Page 4, line 146, “….This could not only result in reduced….” is confusing.
5. Page 6, line 263, “….the lowest possible radiation possible exposure in the shortest possible time….” is confusing.
6. Page 12, line 601, the whole paragraph “….Identifying/classifying results, Identifying cases/cohorts for research studies, Identifying follow-up recommendations, Determining Imaging protocols, Diagnostic Surveillance, and Evaluating the Quality of Radiologic Practice are the biggest applications of NPL in the radiology field.” is not easy to understand.
Author Response
The authors thank the reviewer for the precious work and suggestions.
We revised the whole manuscript, removing paragraph to shorten the text, trying to make it easier to read and rewrote many sentences to make them clear.
We performed the required changes as follows:
- I suggest that the authors can present their review in a more succinct style.
We remove extensive paragraph from the manuscript to fit the suggestion
- There are too many abbreviations in the text and can be misleading or misrepresented especially if they are not checked properly. Some abbreviations are even not spelled out for their first appearance in the text. I strongly recommend that the authors can reduce their frequency.
Thank you for the suggestion.
We removed most abbreviations, using the extended version of the names, as suggested
- Page 3, line 103, “….data, hence this type of ML. This type of ML addresses…” is not easy to understand.
We reformulated the sentence to make the meaning clear
- Page 4, line 146, “….This could not only result in reduced….” is confusing.
We reformulated the sentence to make the meaning clear
- Page 6, line 263, “….the lowest possible radiation possible exposure in the shortest possible time….” is confusing.
We reformulated the sentence to make the meaning clear
- Page 12, line 601, the whole paragraph “….Identifying/classifying results, Identifying cases/cohorts for research studies, Identifying follow-up recommendations, Determining Imaging protocols, Diagnostic Surveillance, and Evaluating the Quality of Radiologic Practice are the biggest applications of NPL in the radiology field.” is not easy to understand.
We reformulated the sentence to make the meaning clear
Thank you very much
Best regards
The authors
Reviewer 2 Report
In this narrative review, the authors present a comprehensive review on the AI applications in the emergency imaging setting.
The article is complete and well presented.
I suggest the following changes:
Carefully revised the whole manuscript, rewriting some phrases that can be difficult to read
Intro
AI …. a structured report. : this sentence should be rewritten in a clear way
Add an explicative figure on supervised learning, and more details on the different types of machine learning
Acquisition
I suggest adding a figure to explain the potential applications of AI in image acquisition/flowchart
Worklist prioritization
“AI-triage triage could reprioritize the worklist by putting studies it considered posi- 294 tive at the top of the list and studies it considers negative at the bottom. Davis et al., for 295 example, demonstrated a significant decrease in report turnaround time, 67 to 59 minutes, 296 and length of stay from 471 to 425 minutes” Triage repeated in the final section
Intracranial hemorrhage and stroke
Is ICH intracranial hemorrhage?
AI can enable the automatic identification of different emergent brain emergencies to speed up the management and plan of the most appropriate therapy .: emergent and emergency are repetitions
...reported, overall sensitivity specificity, and accuracy of … a comma is missing; some other punctuation is missing. Revise this paragraph carefully
…which has been assessed which has assessed, instead of has been
Bone fracture
“This is commonly ascribed to the continual increase in the requested imaging examinations, which places increasing pressure on radiology departments that must deliver prompt responses, and this constantly growing workload results in stress at work and burnout” this sentence should be rewritten in a clearer way.
Abdominal emergencies
“As emergency radiology has to deal with an increasing request for specialistic examinations with reporting times as short as possible, this field represents a promising avenue for AI in radiology departments” this concept has been already stated. You can remove the paragraph
Small bowel occlusion
“(445 with normal findings and 445 with small bowel obstruction” the second “)” is missing
Instussusception
“from patients without intussusception “ replace “from” with “of”
Challenges and perspectives
Add something on the possible validation methods, such as K-fold, boostrapping, etc...
Author Response
The authors thank the reviewers for his/her work and precious suggestions.
We performed all required changes as follows:
In this narrative review, the authors present a comprehensive review on the AI applications in the emergency imaging setting.
The article is complete and well presented.
Thank you for the positive comments
I suggest the following changes:
Carefully revised the whole manuscript, rewriting some phrases that can be difficult to read
We revised the whole manuscript, rewriting many sentences to make them clear and cutting the redundant concepts
Intro
AI …. a structured report. : this sentence should be rewritten in a clear way
We rewrote the sentence in a clear way
Add an explicative figure on supervised learning, and more details on the different types of machine learning
Thank you for the suggestions
We add explicative figures and more details on the different approaches of ML
Acquisition
I suggest adding a figure to explain the potential applications of AI in image acquisition/flowchart
We added a figure as suggested
Worklist prioritization
“AI-triage triage could reprioritize the worklist by putting studies it considered posi- 294 tive at the top of the list and studies it considers negative at the bottom. Davis et al., for 295 example, demonstrated a significant decrease in report turnaround time, 67 to 59 minutes, 296 and length of stay from 471 to 425 minutes” Triage repeated in the final section
We removed the repetition
Intracranial hemorrhage and stroke
Is ICH intracranial hemorrhage?
We eliminated this abbreviation
AI can enable the automatic identification of different emergent brain emergencies to speed up the management and plan of the most appropriate therapy .: emergent and emergency are repetitions
We removed the repetition
...reported, overall sensitivity specificity, and accuracy of … a comma is missing; some other punctuation is missing. Revise this paragraph carefully
We revised the punctuation
…which has been assessed which has assessed, instead of has been
Bone fracture
“This is commonly ascribed to the continual increase in the requested imaging examinations, which places increasing pressure on radiology departments that must deliver prompt responses, and this constantly growing workload results in stress at work and burnout” this sentence should be rewritten in a clearer way.
We rewrote the sentence
Abdominal emergencies
“As emergency radiology has to deal with an increasing request for specialistic examinations with reporting times as short as possible, this field represents a promising avenue for AI in radiology departments” this concept has been already stated. You can remove the paragraph
Thank you, we removed the section to avoid redundancy
Small bowel occlusion
“(445 with normal findings and 445 with small bowel obstruction” the second “)” is missing
We apologize for the mistake and corrected the sentence
Instussusception
“from patients without intussusception “ replace “from” with “of”
We apologize for the mistake and corrected the sentence
Challenges and perspectives
Add something on the possible validation methods, such as K-fold, boostrapping, etc...
We add details on the suggested topic in the appropriate section
Thank you
Best regards
The authors
Reviewer 3 Report
Interesting and well written study regarding the role of artificial intelligence in emergency radiology.
The study summarize all the relevant methodology regarding artificial intelligence, and its utility in common clinical practice.
I think this is worth for publication since AI becomes essential in everyday working.
Just a minor comment regarding stroke: I would suggest to add the utility of some algorithm of AI, such as colorviz, in the diagnosis of arterial intracranial vessels occlusion (you can find some information here Verdolotti T et al. ColorViz, a New and Rapid Tool for Assessing Collateral Circulation during Stroke. Brain Sci. 2020 Nov 20;10(11):882. doi: 10.3390/brainsci10110882. PMID: 33233665; PMCID: PMC7699692.). This is a very powerful tool used by many centers to aid the detection of intracranial vessels occlusion.
Conclusion are concise and well balanced.
Author Response
The authors thank the reviewers for the work and precious suggestions.
We performed all required changes as follows:
Interesting and well written study regarding the role of artificial intelligence in emergency radiology.
The study summarize all the relevant methodology regarding artificial intelligence, and its utility in common clinical practice.
I think this is worth for publication since AI becomes essential in everyday working.
Thank you for the positive comments
Just a minor comment regarding stroke: I would suggest to add the utility of some algorithm of AI, such as colorviz, in the diagnosis of arterial intracranial vessels occlusion (you can find some information here Verdolotti T et al. ColorViz, a New and Rapid Tool for Assessing Collateral Circulation during Stroke. Brain Sci. 2020 Nov 20;10(11):882. doi: 10.3390/brainsci10110882. PMID: 33233665; PMCID: PMC7699692.). This is a very powerful tool used by many centers to aid the detection of intracranial vessels occlusion.
We inserted the above reference in the section about neuroradiology emergencies, as suggested
Conclusion are concise and well balanced.
Thank you for the positive comment.
Thank you
Best regards
The authors